# Association between *IL-8 (-251T/A)* and *IL-6 (-174G/C)* Polymorphisms and Oral Cancer Susceptibility: A Systematic Review and Meta-Analysis

**DOI:** 10.3390/medicina57050405

**Published:** 2021-04-22

**Authors:** Farzad Rezaei, Hady Mohammadi, Mina Heydari, Masoud Sadeghi, Hamid Reza Mozaffari, Atefeh Khavid, Mostafa Godiny, Serge Brand, Kenneth M. Dürsteler, Annette Beatrix Brühl, Dominik Cordier, Dena Sadeghi-Bahmani

**Affiliations:** 1Department of Oral and Maxillofacial Surgery, Kermanshah University of Medical Sciences, Kermanshah 6713954658, Iran; rezaeifarzad63@yahoo.com; 2Department of Oral and Maxillofacial Surgery, Health Services, Kurdistan University of Medical Sciences, Sanandaj 6617713446, Iran; hadi.mohammadi@muk.ac.ir; 3Students Research Committee, Kermanshah University of Medical Sciences, Kermanshah 6715847141, Iran; mina_heydari2@yahoo.com; 4Medical Biology Research Center, Kermanshah University of Medical Sciences, Kermanshah 6714415185, Iran; sadeghi_mbrc@yahoo.com; 5Department of Oral and Maxillofacial Medicine, Kermanshah University of Medical Sciences, Kermanshah 6713954658, Iran; mozaffari20@yahoo.com; 6Department of Oral and Maxillofacial Radiology, Kermanshah University of Medical Sciences, Kermanshah 6713954658, Iran; atefehkhavid@gmail.com; 7Department of Endodontics, Kermanshah University of Medical Sciences, Kermanshah 6713954658, Iran; mostafa_goodin@yahoo.com; 8Sleep Disorders Research Center, Kermanshah University of Medical Sciences, Kermanshah 6719851115, Iran; dena.sadeghibahmani@upk.ch; 9Center for Affective, Stress and Sleep Disorders (ZASS), Psychiatric University Hospital Basel, 4002 Basel, Switzerland; annette.bruehl@upk.ch; 10Department of Clinical Research, University of Basel, 4031 Basel, Switzerland; 11Department of Sport, Exercise and Health, Division of Sport Science and Psychosocial Health, University of Basel, 4052 Basel, Switzerland; 12Substance Abuse Prevention Research Center, Kermanshah University of Medical Sciences, Kermanshah 6715847141, Iran; 13School of Medicine, Tehran University of Medical Sciences, Tehran 1416753955, Iran; 14Psychiatric Clinics, Division of Substance Use Disorders, University of Basel, 4052 Basel, Switzerland; Kenneth.Duersteler@upk.ch; 15Center for Addictive Disorders, Department of Psychiatry, Psychotherapy and Psychosomatics, Psychiatric Hospital, University of Zurich, 8001 Zurich, Switzerland; 16Department of Neurosurgery, University Hospital Basel, 4031 Basel, Switzerland; dominik.cordier@usb.ch; 17Departments of Physical Therapy, University of Alabama at Birmingham, Birmingham, AL 35209, USA

**Keywords:** oral carcinoma, oral cavity cancer, polymorphism, cytokine, interleukin, meta-analysis

## Abstract

Background and objective: Inflammation and cell-mediated immunity can have significant roles in different stages of carcinogenesis. The present meta-analysis aimed to evaluate the association between the polymorphisms of *IL-8 (-251T/A)* and *IL-6 (-174G/C)* and the risk of oral cancer (OC). Methods: PubMed/MEDLINE, Web of Science, Cochrane Library, and Scopus databases were searched until December 18, 2020 without any restrictions. RevMan 5.3 software was used to calculate the results of forest plots (odds ratios (ORs) and 95% confidence intervals (CIs)); CMA 2.0 software was used to calculate funnel plots (Begg’s and Egger’s tests), and SPSS 22.0 was used for the meta-regression analysis. Moreover, trial sequential analysis was conducted to estimate the robustness of the results. Results: Eleven articles including twelve studies were selected for the meta-analysis. The pooled ORs for the association between *IL-8 (-251T/A)* polymorphism and the risk of OC in the models of A vs. T, AA vs. TT, TA vs. TT, AA + TA vs. TT, and AA vs. TT + TA were 0.97 (*p* = 0.78), 0.86 (*p* = 0.55), 0.78 (*p* = 0.37), 0.83 (*p* = 0.45), and 1.10 (*p* = 0.34), respectively. The pooled ORs *IL-6 (-174G/C)* polymorphism and the risk of OC in the models of C vs. G, CC vs. GG, GC vs. GG, CC + GC vs. GG, and CC vs. GG + GC were 1.07 (*p* = 0.87), 1.17 (*p* = 0.82), 1.44 (*p* = 0.38), 1.28 (*p* = 0.61), and 0.96 (*p* = 0.93), respectively. There was no association between *IL-8 (-251T/A)* polymorphism and OC susceptibility, but the C allele and GC and CC genotypes of *IL-6 (-174G/C)* polymorphism were associated with the risk of OC based on subgroup analyses, that is to say, the source of control and the genotyping method might bias the pattern of association. Conclusions: The meta-analysis confirmed that there was no association between the polymorphisms of *IL-6 (-174G/C)* and *IL-8 (-251T/A)* and the susceptibility of OC. However, the source of control and the genotyping method could unfavorably impact on the association between the polymorphisms of *IL-6 (-174G/C)* and the risk OC.

## 1. Introduction

Oral cancer (OC) is the 11th most common malignancy in the world. The incidence and mortality of this malignancy varies according to geographical conditions [1]. Thus, this cancer shows a wide variation in distribution among countries and geographical areas [2]. In 2018, the last year for which the International Agency for Research on Cancer (IARC) data are available, the global age-standardized risk for OC was 5.2 for males and 2.3 for females [3]. The estimated incident cases of OC globally elevated from 185,976 cases in 1990 to 389,760 cases in 2017 and an increase in deaths from 97,492 deaths in 1990 to 193,696 deaths in 2017 [4]. The most malignant neoplasm (more than 90%) of OC is the oral squamous cell carcinoma (OSCC), which causes damage to the epithelial cells of the mouth area as a result of the accumulation of multiple genetic mutations in the cells [5,6,7]. Higher age, male sex, and adverse socioeconomic conditions are common risk factors for this cancer [8,9]. Additional risk factors for OSSC are: tobacco smoking, use of smokeless tobacco products [10,11], chewing of betel quid [12], viral factors such as human papillomavirus [13], ultraviolet light [9], periodontal disease, infections, alcohol consumption, poor oral hygiene, and diet with low Mediterranean-like fruit and vegetables [14]. Furthermore, the early detection of oral tumors has not improved over time, and up to 77% of cases of this cancer were diagnosed in advanced stages [15]. Next, conventional treatments for this cancer include surgery, radiotherapy, and chemotherapy [16].

Both genetic factors and environmental carcinogens were associated with the risk of OC [17]. Altered genetic abnormalities of carcinogenic metabolism, DNA repair, and cell cycle were identified as possible mediators of oral tumorigenesis [18,19]. Furthermore, inflammation and cell-mediated immunity can have important functions at different stages of carcinogenesis [20]. In this view, there were two main cytokines (Interleukin (IL)-6 and IL-8) related to inflammation in several diseases. The *IL-8* gene is located on chromosome 4q13-3 in a proximal region of the promoter [21], and a wide range of cell types, such as neutrophils, macrophages, endothelial cells, and epithelial cells, produce IL-8 [22]. The *IL-6* gene is located at 7p21.24; the *IL-6 (-174G/C)* polymorphism is associated with the 5 ‘UTR region containing the promoter and affects their transcripts as well as its serum levels [23]. The results of a meta-analysis showed that salivary and serum levels of IL-6 and IL-8 in individuals with OSCC were significantly higher than the salivary and serum levels of IL-6 and IL-8 in healthy controls [24]. In 2013, a meta-analysis with six studies on the association between the *IL-8 (-251T/A)* polymorphism and the risk of OC showed that this polymorphism may increase the risk of OC, especially among European populations [25]; one year later, a further meta-analysis with similar studies confirmed the previous pattern of results [26]. Given this background, it appeared that some polymorphisms of cytokines may increase the risk of OC [27,28,29,30,31]. As an overall result, it appeared that cytokines, including their polymorphisms, were associated with the risk of OC. In contrast, there is no meta-analysis on the association between *IL-6* polymorphism and the susceptibility of OC. The aim of the present meta-analysis was examining the role of the two most important cytokines, namely *IL-8 (-251T/A)* and *IL-6 (-174G/C)*, and their polymorphisms on the risk of OC, including more studies in this field, in contrast to previous meta-analyses.

## 2. Materials and Methods

The approval of an ethics committee was not required, because data were extracted from secondary data. This systematic review was performed according to the Preferred Reporting Items for Systematic Reviews and Meta-Analyses (PRISMA) protocols [32]. We formulated the following PICO (participants of interest, intervention, control, and outcome of interest) question: Are *IL-8 (-251T/A)* and *IL-6 (-174G/C*) polymorphisms associated with the OC risk comparing the prevalence of their alleles and genotypes in OC patients compared to controls?

### 2.1. Data Sources and Literature Search

A systematic electronic search was comprehensively performed in PubMed/MEDLINE, Web of Science, Cochrane Library, and Scopus databases until December 18, 2020 without restrictions. The used search terms were (“interleukin-8” or “IL-8” or “IL8” or “interleukin-6” or “IL-6” or “IL6”), (“oral cancer*” or “oral carcinoma*” or “oral cavity cancer*” or “oral cavity carcinoma*” or “oral squamous cell carcinoma*” or “oral SCC” or “OSCC” or “tongue cancer*” or “tongue carcinoma*” or “oropharyngeal squamous cell carcinoma*” or “oropharynx cancer*” or “oropharynx carcinoma*” or “oropharyngeal cancer*” or “oropharyngeal carcinoma*” or “oropharyngeal neoplasm*” or “oropharynx neoplasm*” or “mouth neoplasm*” or “mouth cancer*” “mouth tumor*” or “oral neoplasm*” or “salivary gland cancer*” or “salivary gland tumor*” or “lip cancer*” or “lip carcinoma*”), and (“polymorphism*” or “variant*” or “allele*” or “genotype*”). An independent review of titles and abstracts was conducted by two reviewers (F.R. and M.S.). Disagreements were resolved by consensus with a third author (S.B.). Other databases and websites were manually checked for relevant studies, and we also checked the references of all subject-related studies that followed the criteria so that no study was missed.

### 2.2. Eligibility Criteria and Study Selection

Inclusion criteria were: (1) studies with a case-control design focused on the associations between *IL-8 (-251A/T)* or *IL-6 (-174G/C)* polymorphisms and the risk of OC; (2) pathological or histological examinations confirmed OC; (3) studies reporting the frequencies of alleles or genotypes; (4) human studies; (5) studies with/without a deviation of the Hardy–Weinberg equilibrium (HWE) for the control group. Exclusion criteria were: (1) duplicate publications; (2) animal studies; (3) reviews, meta-analyses, and conference papers; (4) studies without control group. For duplicate publications, we selected the one with the newest date. One author checked full-text papers based on the criteria (M.S.). An independent review of full-text papers was conducted by two reviewers (F.R. and M.S.) and disagreements were resolved by discussion between both reviewers. Agreement was assessed using the Kappa statistic as defined in the Cochrane Handbook [33]. The Kappa statistic was calculated using GraphPad software (https://www.graphpad.com/quickcalcs/kappa1/, accessed on 5 January 2021). Kappa statistic values were interpreted as: K = 0.40–0.59 (Fair agreement), K = 0.60–0.74 (Good agreement), and K = 0.75 or more (Excellent agreement).

### 2.3. Data Extraction

The data from published studies were extracted independently by two reviewers (M.H. and H.M.) to retrieve the necessary information. In case of discrepancies between the data of the two previous reviewers, the review was performed by a separate reviewer (D.S.B., K.M.D. and D.K.).

### 2.4. Quality Assessment

Two reviewers (F.R. and M.S.) independently assessed the quality of the selected studies by scoring them according to a set of pre-established criteria based on Table 1 in the study of Yang et al. [26], and disagreements were resolved by a short discussion. The range of scores varies from 0 to 12, with higher scores indicating better study quality.

### 2.5. Statistical Analysis

The association between polymorphisms and the OC susceptibility was estimated by odds ratios (ORs) with 95% confidence intervals (CIs). The used genotype models for *IL-8* and *IL-6* polymorphisms were (allele: A vs. T and C vs. G), (homozygote: AA vs. TT and CC vs. GG), (heterozygote: TA vs. TT and GC vs. GG), (recessive: AA + TA vs. TT and CC + GC vs. GG), and (dominant: AA vs. TT + TA and CC vs. GG + GC). To estimate heterogeneity, a chi-square-based Q test and inconsistency index I^2^ were used among the studies [34,35], where a *p*-value > 0.10 on the Q test and I^2^ < 50% identified that there was no heterogeneity among the studies. While there was heterogeneity, the pooled OR was estimated by the random-effects model [36]; otherwise, we used the fixed-effects model [37]. Subgroup analysis is an analysis method that is performed by breaking study samples into smaller subsets based on a common feature, and the goal is to explore the effects of different factors on the results. Meta-regression is another quantitative method used in meta-analyses to estimate the effect of confounding variables on initial results, including variables such as year of publication and number of participants. Funnel plots were constructed to check whether the publication bias might affect the validity of the estimates. The diagnosis of asymmetry of funnels was performed using Begg’s and Egger’s tests, which are linear regression methods for measuring the symmetry of funnels. Asymmetry can be a reason for bias in studies; hence, *p*-values < 0.05 were chosen for the tests. The *p*-values (two-sided) < 0.05 were considered to show significance, unless specifically mentioned. The results of the forest plots were obtained by Review Manager 5.3 (RevMan 5.3) software, funnel plots by Comprehensive Meta-Analysis version 2.0 (CMA 2.0) software, and meta-regression by SPSS 22.0 software.

Meta-analysis may cause a false-positive or negative conclusion [38]. Hence, we applied trial sequential analysis (TSA) by using TSA software (version 0.9.5.10 beta) (Copenhagen Trial Unit, Centre for Clinical Intervention Research, Rigshospitalet, Copenhagen, Denmark) to decrease these statistical errors [39]. Additionally, a threshold of futility could be examined by TSA to find a conclusion of no effect before reaching the information size. We calculated the required information size (RIS) based on an alpha risk of 5%, a beta risk of 20%, and a two-sided boundary type. For those analyses where the Z-curve reached the RIS line or monitored the boundary line or futility area, enough samples are involved in the studies, and their results are valid. Otherwise, the amount of information is not large enough, and more evidence is needed.

## 3. Results

### 3.1. Study Selection

To search four main databases and other sources, 94 records were retrieved (Figure 1). After removing duplicates and irrelevant records, 20 full-text articles evaluated for eligibility; 9 full-texts papers were excluded with reasons (2 reviews, 2 meta-analyses, 1 animal study, 1 had no control group, 1 conference paper, and 2 duplicate publications). Accordingly, 11 articles were selected for the meta-analysis.

### 3.2. Full Text Evaluation

A total of 20 full-text papers were evaluated for eligibility and the reviewer agreement was computed using kappa scores, and was found to be excellent at 0.818 (95% Confidence Interval: 0.601 to 1.000) (Table 1).

### 3.3. Study Characteristics

The characteristics of the articles are shown in Table 2. Of the 11 articles included, 7 studies [30,40,41,42,43,44,45] reported *IL-8 (-251T/A)* polymorphism, 3 [27,28,46] reported *IL-6 (-174G/C)* polymorphism, and 1 reported [31] both polymorphisms. Six articles [27,28,30,31,40,46] reported Caucasian participants, four studies [42,43,44,45] reported Asian participants, and one study [41] reported mixed ethnicities. Eight articles [30,31,40,41,43,44,45,46] had population-based controls, while three articles [27,28,42] had hospital-based controls. Ten out of eleven articles had individuals with OSCC; one article [45] had patients with tongue SCC.

The genotyping method was TaqMan in four articles [27,40,41,43]; polymerase chain reaction-restriction fragment length polymorphism (PCR-RFLP) was used in five studies [28,30,31,44,46], while two studies [42,45] used other PCRs.

### 3.4. Meta-Analysis

The distribution of alleles and genotypes of *IL-8 (-251T/A)* and *IL-6 (-174G/C)* polymorphisms, the quality of the selected studies, and the *p*-values of HWE are shown in Table 3. The controls in two articles [30,31] had a deviation of HWE (*p* < 0.001). In addition, the quality of the selected studies is shown in Table 3.

The forest plot analysis of the associations between *IL-8 (-251T/A)* polymorphism and susceptibility to OC based on five genetic models is shown in Table 4. The pooled ORs for the models of A vs. T, AA vs. TT, TA vs. TT, AA + TA vs. TT, and AA vs. TT + TA were 0.97 [95%CI: 0.76, 1.23; *p* = 0.78; I^2^ = 78% (P_h_ < 0.0001)], 0.86 [95%CI: 0.53, 1.41; *p* = 0.55; I^2^ = 71% (P_h_ = 0.001)], 0.78 [95%CI: 0.46, 1.33; *p* = 0.37; I^2^ = 88% (P_h_ < 0.00001)], 0.83 [95%CI: 0.51, 1.35; *p* = 0.45; I^2^ = 88% (P_h_ < 0.00001)], and 1.10 [95%CI: 0.90, 1.33; *p* = 0.34; I^2^ = 38% (P_h_ = 0.13)], respectively. There was no association between *IL-8 (-251T/A)* polymorphism and susceptibility to OC.

The forest plot analysis of the associations between *IL-6 (-174G/C)* polymorphism and the susceptibility to OC based on five genetic models is shown in Table 5. The pooled ORs for the models of C vs. G, CC vs. GG, GC vs. GG, CC + GC vs. GG, and CC vs. GG + GC were 1.07 [95%CI: 0.50, 2.26; *p* = 0.87; I^2^ = 93% (P_h_ < 0.00001)], 1.17 [95%CI: 0.31, 4.36; *p* = 0.82; I^2^ = 87% (P_h_ < 0.0001)], 1.44 [95%CI: 0.64, 3.26; *p* = 0.38; I^2^ = 88% (P_h_ < 0.0001)], 1.28 [95%CI: 0.50, 3.26; *p* = 0.61; I^2^ = 92% (P_h_ < 0.00001)], and 0.96 [95%CI: 0.37, 2.50; *p* = 0.93; I^2^ = 81% (P_h_ = 0.001)], respectively. There was no association between *IL-6 (-174G/C)* polymorphism and susceptibility to OC.

Table 6 shows the subgroup analysis based on the ethnicity, source of control, and genotyping method for the association between the *IL-8 (-251T/A)* polymorphism and the susceptibility to OC. There was no association between this polymorphism and the susceptibility to OC.

### 3.5. Subgroup Analysis

Table 7 shows the subgroup analysis based on the ethnicity, source of control, and genotype method for the association between *IL-6 (-174G/C)* polymorphism and the risk of OC. The source of control and the genotyping method were influencing factors on the association. The C allele and GC genotype had an elevated risk of OC in the studies with population-based controls, and the C allele and CC genotype had a reduced risk of OC in the studies with hospital-based controls. In addition, the C allele had a reduced risk of OC, and the CC genotype had an elevated risk of OC.

### 3.6. Meta-Regression

Table 8 shows the meta-regression analysis based on the year of publication and number of participants for the association between *IL-8 (-251T/A)* and *IL-6 (-174G/C)* polymorphisms and the susceptibility to OC. The year of publication and number of participants were not confounding factors on the association.

### 3.7. Sensitivity Analysis

The sensitivity analyses, namely the “cumulative analysis” and “one study removed,” showed the consistency/stability of the results. We deleted two studies [30,31] reporting *IL-8 (-251T/A)* polymorphism with a deviation of HWE for control group; the pooled OR changed to 0.96 [95%CI: 0.84, 1.09; *p* = 0.53; I^2^ = 0% (P_h_ = 0.54)], 0.95 [95%CI: 0.72, 1.24; *p* = 0.69; I^2^ = 0% (P_h_ = 0.65)], 0.91 [95%CI: 0.74, 1.12; *p* = 0.37; I^2^ = 0% (P_h_ = 0.43)], 0.91 [95%CI: 0.75, 1.11; *p* = 0.37; I^2^ = 0% (P_h_ = 0.56)], and 0.99 [95%CI: 0.78, 1.26; *p* = 0.96; I^2^ = 3% (P_h_ = 0.40)] for allele, homozygote, heterozygote, recessive, and dominant models, respectively. The new results confirmed the initial results with a lack of heterogeneity. Removing one study with outlier data [31], the pooled OR for *IL-8 (-251T/A)* polymorphism became 0.87 [95%CI: 0.73, 1.04; *p* = 0.14; I^2^ = 53% (P_h_ = 0.05)], 0.77 [95%CI: 0.51, 1.16; *p* = 0.21; I^2^ = 61% (P_h_ = 0.02)], 0.68 [95%CI: 0.40, 1.17; p = 0.16; I2 = 86% (P_h_ < 0.00001)], 0.71 [95%CI: 0.45, 1.13; *p* = 0.15; I^2^ = 83% (P_h_ < 0.00001)], and 1.03 [95%CI: 0.84, 1.25; *p* = 0.80; I^2^ = 0% (P_h_ = 0.50)] for allele, homozygote, heterozygote, recessive, and dominant models, respectively. The new pooled ORs had no significant difference with the initial pooled ORs.

### 3.8. Publication Bias

The results of Egger’s and Begg’s tests for allele, homozygote, heterozygote, recessive, and dominant models were (*p* = 0.09841 and *p* = 0.13756), (*p* = 0.25251 and *p* = 0.32230), (*p* = 0.88968 and *p* = 0.45790), (*p* = 0.93569 and *p* = 0.45790), and (*p* = 0.40111 and *p* = 0.45790) for *IL-8 (-251T/A)* polymorphism; and (*p* = 0.15449 and *p* = 0.49691), (*p* = 0.74211 and *p* =1.00000), (*p* = 0.62032 and *p* = 0.49691), (*p* = 0.57329 and *p* = 0.49961), and (*p* = 0.52563 and *p* = 0.17423) for *IL-6 (-174G/C)* polymorphism, respectively (Figure 2). Therefore, the results of the tests did not reveal any publication bias across and between the studies.

### 3.9. Trial Sequential Analysis

In the study of the *IL-8 (-251T/A)* polymorphism, the Z-curve of the allele, homozygote, heterozygote, and recessive models reached the futility area, confirming that the *IL-8 (-251T/A)* polymorphism was not associated with the OC risk. With regards to *IL-6 (-174G/C)* polymorphism, the Z-curve of the allele, heterozygote, recessive, and dominant models reached futility area, confirming that the *IL-6 (-174G/C)* polymorphism was not associated with the OC risk (Figure 3).

## 4. Discussion

The main findings of the present systematic review, meta-analysis, and meta-regression were that there was no association between *IL-8 (-251T/A)* and *IL-6 (-174G/C)* polymorphisms and susceptibility to OC; the TSA confirmed this result. The ethnicity, source of control, and genotyping method were not confounding factors on the association between *IL-8 (-251T/A)* polymorphism and the susceptibility to OC; in contrast, the source of control and genotyping method were influencing factors on the association between *IL-6 (-174G/C)* polymorphism and the risk of OC. In addition, based on meta-regression analysis, the year of publication and number of participants were not confounding factors on this association. The funnel plots did not reveal any publication bias across and between the included studies in the meta-analysis.

Inflammation is an important factor in the pathogenesis of human cancer [47,48] in general and for OC specifically [49]. Polymorphisms in the promoter region or other regulatory regions of the cytokine gene may impact cytokine expression [23], such as *IL-8 (-251T/A)* and *IL-6 (-174G/C)* polymorphisms. Some studies were unable to find an association between *IL-8 (-251T/A)* [40,41,42,43,44,45] and the susceptibility to OC; in contrast, other studies found a significant association, including a protective role [30] or elevated risk [31] of this polymorphism in the development of OC. In addition, studies reporting *IL-6 (-174G/C)* polymorphism showed a protective role [27,28] or elevated risk [31,46] of this polymorphism in the development of OC. Our meta-analysis showed a lack of association between these polymorphisms (*IL-6 (-174G/C)* and *IL-8 (-251T/A)*) with the risk of OC. Confounders such as age, sex, ethnicity, source of controls, genotyping method, the year of publication, number of participants, and environmental factors, may explain the contradictory findings between the present and previous results. In a similar vein, and based on the subgroup analysis, we found that the source of control and genotyping method were influencing factors on the association between *IL-6 (-174G/C)* polymorphism and the risk of OC.

One study [31] reported that the homozygous *IL-6 (-174G/C)* polymorphism was significantly associated with both overall OSCC stages and the early and advanced OSCC stages. In contrast, *IL-8 (-251T/A)* polymorphism was significantly correlated with overall and early OSCC stages. In addition, Vairaktaris et al. [50] observed that the C allele of *IL-6 (-174G/C)* polymorphism had a higher risk of OC in high stages than it in low stages. Singh et al. [46] reported that *IL-6 (-174G/C)* polymorphism was not associated with tobacco chewing, smoking, and alcohol consumption and the OSCC development; in contrast, Vairaktaris et al. [50] showed an association between this polymorphism and the risk of OSCC in individuals consuming alcohol. However, gene interactions and other environmental factors were not related to OSCC pathogenesis [50]. Singh et al. [30] concluded that there was relationship between *IL-8 (-251T/A)* polymorphism and the clinicopathological status of OC, its related pain. Furthermore, the association between *IL-8 (-251T/A)* polymorphism and other cancers, such as melanoma [51], hepatocellular carcinoma [52], ovarian cancer [53], and breast cancer [54], has been confirmed. However, Liu et al. [44] rejected the role of clinicopathological parameters on the association between *IL-8 (-251T/A)* polymorphism and the susceptibility to OC. Next, environmental factors such as smoking and drinking could impact on the association between *IL-8 (-251T/A)* polymorphism and susceptibility to OSCC, at least among Thai participants [43]. Moreover, compared to individuals with homozygote genotypes, lymph node metastasis were statistically significantly more prevalent in participants with a heterozygote genotype of *IL-8 (-251T/A)* polymorphism. Therefore, the role of clinicopathological and environmental factors on the association between both polymorphisms of *IL-6 (-174G/C)* and *IL-8 (-251T/A)* should be considered in future studies.

The limitations of the present work were: (1) The small number of published studies on these topics and associations; (2) Clinicopathological and environmental factors between two groups (cases and controls) were not reported in the studies; (3) Different genotyping methods might have biased the pattern of results; (4) The small number of participants in some studies. However, the limitations should be balanced against the following strength: (1) The lack of publication bias; (2) The high quality of the studies.

## 5. Conclusions

The finding of this systematic review, meta-analysis, and meta-regression showed that there was no association between the polymorphisms of *IL-6 (-174G/C)* and *IL-8 (-251T/A)* and the susceptibility to OC. However, the source of control and the genotyping method could impact the association of the polymorphisms of *IL-6 (-174G/C)* with the risk of OC. We also observed contradictory results between the present and previous patterns of results. Furthermore, possible confounders, such as tobacco smoking, use of smokeless tobacco products, chewing of betel quid, viral factors such as human papillomavirus, ultraviolet light, periodontal disease, infections, alcohol consumption, poor oral hygiene, diet with low Mediterranean-like fruit and vegetables, and adverse socioeconomic conditions should be thoroughly assessed and introduced as mediating factors between the interplay of these polymorphisms and the risk of oral cancer.

## Figures and Tables

**Figure 1 medicina-57-00405-f001:**
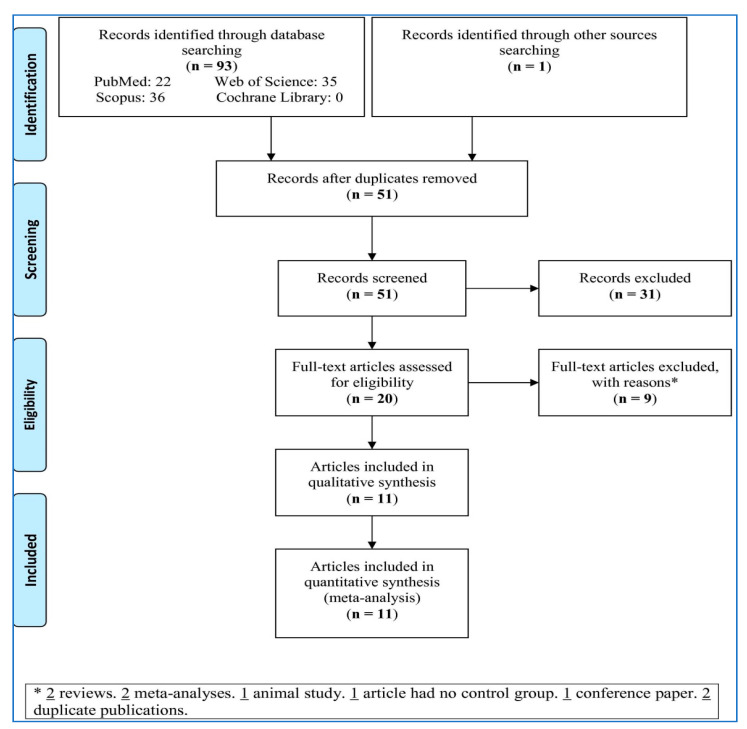
Flowchart of the meta-analysis.

**Figure 2 medicina-57-00405-f002:**
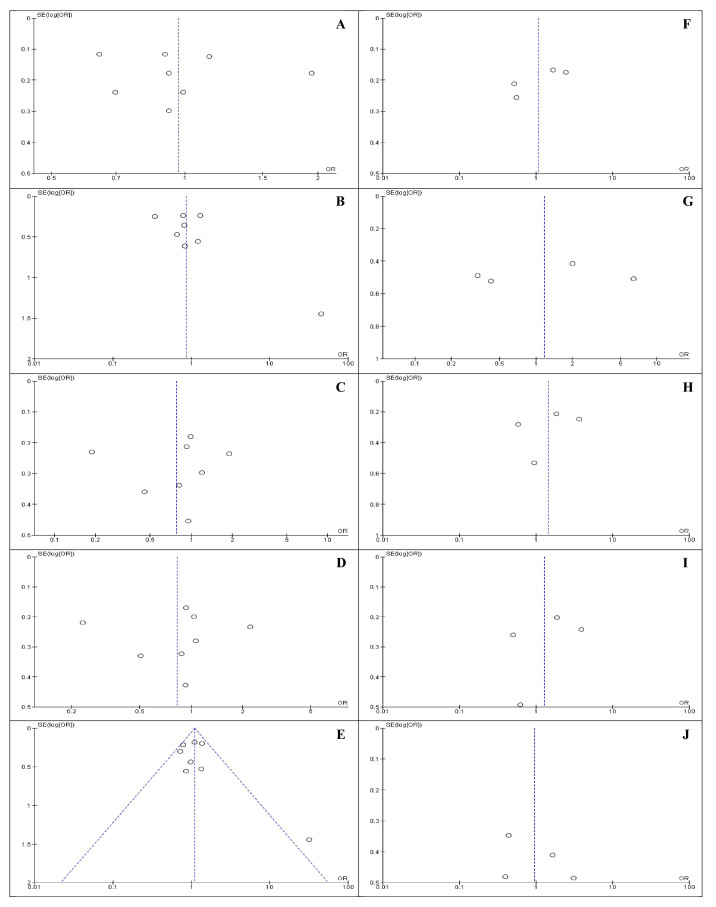
Funnel plot of the associations between *IL-8 (-251T/A)* and *IL-6 (-174G/C)* polymorphisms (**A**–**E**) for IL-8 and (**F**–**J**) for IL-6 show allele, homozygote, heterozygote, recessive, and dominant models, respectively). Each point illustrates a separate study for the association. SE (Log [OR]), Standard error (natural logarithm of odds ratio [OR]). Horizontal line, mean magnitude of the effect. Note: Funnel plot with pseudo 95% confidence intervals (CIs) was used.

**Figure 3 medicina-57-00405-f003:**
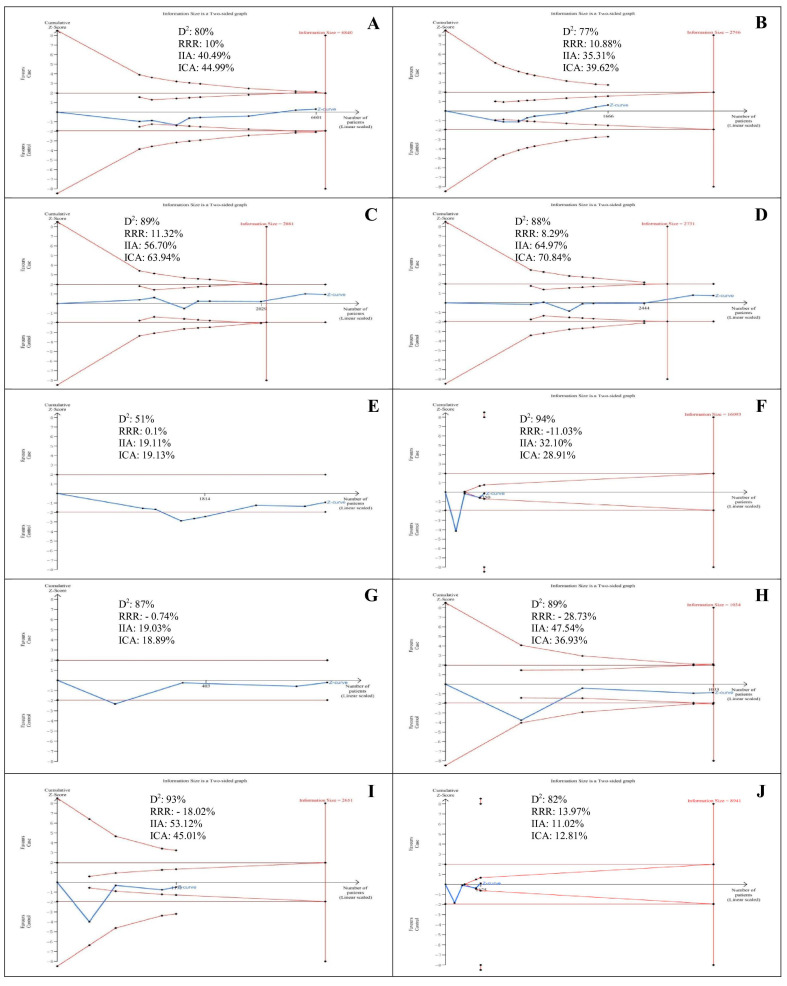
Trial sequential analyses for *IL-8 (-251T/A)* and *IL-6 (-174G/C)* polymorphisms and oral cancer risk *IL-8 (-251T/A)* polymorphism: (**A**–**E**) for allele, homozygote, heterozygote, recessive, and dominant models. *IL-6 (-174G/C)* polymorphism: (**F**–**J**) for allele, homozygote, heterozygote, recessive, and dominant models. Abbreviation: D2, diversity; RRR, relative risk reduction; IIA, incidence in intervention arm; ICA, incidence in control arm. IIA and ICA were calculated from the average incidence in case and control groups. Error α and 1-β were defined as 5% and 80%, respectively, in each model.

**Table 1 medicina-57-00405-t001:** Inclusion and exclusion of full-text papers for the initial search.

Review Author 2 (F.R.)	Review Author 1 (M.S.)				
		**Include**	**Exclude**	**Unsure**	**Total**
	Include	9	0	0	9
	Exclude	0	9	0	9
	Unsure	2	0	0	2
	Total	11	9	0	20

**Table 2 medicina-57-00405-t002:** Characteristics of all articles included in meta-analysis.

First Name, Publication Year	Country	Ethnicity	Source of Controls	Type of Cancer	Genotyping Method	Polymorphism
Campa, 2007 [40]	Central/Eastern Europe	Caucasian	Population-based	Oral SCC	TaqMan	*IL-8 (-251T/A)*
Shimizu, 2008 [45]	Japan	Asian	Population-based	Tongue SCC	PCR-FRET	*IL-8 (- 251T/A)*
Vairaktaris, 2008 [31]	Greece	Caucasian	Population-based	Oral SCC	PCR-RFLP	*IL-8 (-251T/A)* & *IL-6 (-174G/C)*
Kietthubthew, 2010 [43]	Thailand	Asian	Population-based	Oral SCC	TaqMan	*IL-8 (-251T/A)*
Gaur, 2011 [28]	India	Caucasian	Hospital-based	Oral SCC	PCR-RFLP	*IL-6 (-174G/C)*
Hu, 2012 [42]	China	Asian	Hospital-based	Oral SCC	PCR-HRM	*IL-8 (-251T/A)*
Liu, 2012 [44]	Taiwan	Asian	Population-based	Oral SCC	PCR-RFLP	*IL-8 (-251T/A)*
Singh, 2015 [46]	India	Caucasian	Population-based	Oral SCC	PCR-RFLP	*IL-6 (-174G/C)*
Singh, 2016 [30]	India	Caucasian	Population-based	Oral SCC	PCR-RFLP	*IL-8 (-251T/A)*
de Matos, 2019 [41]	Brazil	Mixed	Population-based	Oral SCC	TaqMan	*IL-8 (-251T/A)*
Fernández-Mateos, 2019 [27]	Spain	Caucasian	Hospital-based	Oral SCC	TaqMan	*IL-6 (-174G/C)*

Abbreviations: SCC, squamous cell carcinoma; PCR, polymerase chain reaction; RFLP, restriction fragment length polymorphism; HRM, high resolution melt; FRET, fluorescence resonance energy transfer.

**Table 3 medicina-57-00405-t003:** The allele and genotype distribution of *IL-8 (-251T/A)* and *IL-6 (-174G/C)* polymorphisms.

**First Name, Publication Year**	**Polymorphism**	**Case**						**Control**						***p*-Value** **of HWE**	**Quality Score**
		**T**	**A**	**TT**	**TA**	**AA**	**MAF**	**T**	**A**	**TT**	**TA**	**AA**	**MAF**		
Campa, 2007 [40]	*IL-8 (-251T/A)*	152	154	40	72	41	0.50	950	846	241	468	189	0.47	0.169	11
Shimizu, 2008 [45]	*IL-8 (-251T/A)*	92	46	31	30	8	0.33	121	61	38	45	8	0.34	0.295	9
Vairaktaris, 2008 [31]	*IL-8 (-251T/A)*	200	116	54	88	14	0.37	240	72	84	72	0	0.23	<0.001	9
Kietthubthew, 2010 [43]	*IL-8 (-251T/A)*	85	41	32	21	10	0.32	117	81	34	49	16	0.41	0.813	10
Hu, 2012 [42]	*IL-8 (-251T/A)*	135	83	42	51	16	0.38	36	24	11	14	5	0.40	0.879	9
Liu, 2012 [44]	*IL-8 (-251T/A)*	325	215	97	131	42	0.40	404	296	120	164	66	0.42	0.454	10
Singh, 2016 [30]	*IL-8 (-251T/A)*	323	277	106	111	83	0.46	257	343	34	189	77	0.57	<0.001	9
de Matos, 2019 [41]	*IL-8 (-251T/A)*	135	115	34	67	24	0.46	135	125	37	61	32	0.48	0.492	9
**First Name, Publication Year**	**Polymorphism**	**Case**						**Control**						***p*-Value of HWE**	**Quality Score**
		**G**	**C**	**GG**	**GC**	**CC**	**MAF**	**G**	**C**	**GG**	**GC**	**CC**	**MAF**		
Vairaktaris, 2008 [31]	*IL-6 (-174G/C)*	186	138	42	102	18	0.43	240	72	90	60	6	0.23	0.297	9
Gaur, 2011 [28]	*IL-6 (-174G/C)*	231	49	98	35	7	0.18	171	69	65	41	14	0.29	0.069	8
Singh, 2015 [46]	*IL-6 (-174G/C)*	401	143	150	101	21	0.26	305	65	129	47	9	0.18	0.094	10
Fernández-Mateos, 2019 [27]	*IL-6 (-174G/C)*	57	83	12	33	25	0.59	39	101	8	23	39	0.72	0.126	11

Abbreviations: IL, Interleukin; MAF, minor allele frequency; HWE, Hardy–Weinberg equilibrium.

**Table 4 medicina-57-00405-t004:** Forest plot analysis of the association between *IL-8 (-251T/A)* polymorphism and oral cancer risk based on five genetic models.

Genetic Model	First Author, Publication Year	Case		Control		Weight	Odds Ratio
		Events	Total	Events	Total		M-H, Random, 95%CI
A vs. T	Campa, 2007	154	306	846	1795	14.7%	1.14 [0.89, 1.45]
	Vairaktaris, 2008	116	316	72	312	12.8%	1.93 [1.36, 2.74]
	Shimizu, 2008	46	138	61	182	10.6%	0.99 [0.62, 1.59]
	Kietthubthew, 2010	41	126	81	198	10.6%	0.70 [0.44, 1.11]
	Hu, 2012	83	218	24	60	8.7%	0.92 [0.51, 1.65]
	Liu, 2012	215	540	296	700	15.0%	0.90 [0.72, 1.13]
	Singh, 2016	277	600	343	600	15.0%	0.64 [0.51, 0.81]
	de Matos, 2019	115	250	125	260	12.8%	0.92 [0.65, 1.30]
Subtotal (95%CI)			2494		4107	100.0%	0.97 [0.76, 1.23]
Total events		1047		1848			
Heterogeneity: Tau² = 0.09; Chi² = 31.24, df = 7 (*p* < 0.0001); I² = 78%							Test for overall effect: Z = 0.28 (*p* = 0.78)
AA vs. TT	Campa, 2007	41	81	189	430	17.2%	1.31 [0.81, 2.10]
	Shimizu, 2008	8	39	8	46	10.3%	1.23 [0.41, 3.64]
	Vairaktaris, 2008	14	68	0	84	2.6%	44.96 [2.63, 769.34]
	Kietthubthew, 2010	10	42	16	50	11.9%	0.66 [0.26, 1.68]
	Hu, 2012	16	58	5	16	9.2%	0.84 [0.25, 2.79]
	Liu, 2012	42	139	66	186	17.3%	0.79 [0.49, 1.26]
	Singh, 2016	83	189	77	111	17.0%	0.35 [0.21, 0.57]
	de Matos, 2019	24	58	32	69	14.5%	0.82 [0.40, 1.65]
Subtotal (95%CI)			674		992	100.0%	0.86 [0.53, 1.41]
Total events		238		393			
Heterogeneity: Tau² = 0.31; Chi² = 24.02, df = 7 (*p* = 0.001); I² = 71%							Test for overall effect: Z = 0.59 (*p* = 0.55)
TA vs. TT	Campa, 2007	72	112	468	709	13.4%	0.93 [0.61, 1.41]
	Shimizu, 2008	30	61	45	83	11.9%	0.82 [0.42, 1.58]
	Vairaktaris, 2008	88	142	72	156	13.2%	1.90 [1.20, 3.02]
	Kietthubthew, 2010	21	53	49	83	11.7%	0.46 [0.23, 0.92]
	Liu, 2012	131	228	164	284	13.7%	0.99 [0.69, 1.41]
	Hu, 2012	51	93	14	25	10.4%	0.95 [0.39, 2.32]
	Singh, 2016	111	217	189	223	13.2%	0.19 [0.12, 0.30]
	de Matos, 2019	67	101	61	98	12.5%	1.20 [0.67, 2.14]
Subtotal (95% CI)			1007		1661	100.0%	0.78 [0.46, 1.33]
Total events		571		1062			
Heterogeneity: Tau² = 0.51; Chi² = 59.21, df = 7 (*p* < 0.00001); I² = 88%							Test for overall effect: Z = 0.91 (*p* = 0.37)
AA + TA vs. TT	Campa, 2007	113	153	657	898	13.5%	1.04 [0.70, 1.53]
	Shimizu, 2008	38	69	53	91	11.9%	0.88 [0.47, 1.65]
	Vairaktaris, 2008	102	156	72	156	13.1%	2.20 [1.40, 3.48]
	Kietthubthew, 2010	31	63	65	99	11.8%	0.51 [0.27, 0.97]
	Liu, 2012	173	270	230	350	13.8%	0.93 [0.67, 1.30]
	Hu, 2012	67	109	19	30	10.3%	0.92 [0.40, 2.13]
	Singh, 2016	194	300	266	300	13.2%	0.23 [0.15, 0.36]
	de Matos, 2019	91	125	93	130	12.5%	1.06 [0.62, 1.84]
Subtotal (95%CI)			1245		2054	100.0%	0.83 [0.51, 1.35]
Total events		809		1455			
Heterogeneity: Tau² = 0.43; Chi² = 56.11, df = 7 (*p* < 0.00001); I² = 88%							Test for overall effect: Z = 0.76 (*p* = 0.45)
AA vs. TT + TA	Campa, 2007	41	153	189	898	20.8%	1.37 [0.93, 2.03]
	Vairaktaris, 2008	14	156	0	156	0.2%	31.85 [1.88, 538.79]
	Shimizu, 2008	8	69	8	91	3.2%	1.36 [0.48, 3.83]
	Kietthubthew, 2010	10	63	16	99	5.4%	0.98 [0.41, 2.32]
	Liu, 2012	42	270	66	350	25.1%	0.79 [0.52, 1.21]
	Hu, 2012	16	109	5	30	3.5%	0.86 [0.29, 2.58]
	Singh, 2016	83	300	77	300	28.8%	1.11 [0.77, 1.59]
	de Matos, 2019	24	125	32	130	13.1%	0.73 [0.40, 1.32]
Subtotal (95%CI)			1245		2054	100.0%	1.10 [0.90, 1.33]
Total events		238		393			
Heterogeneity: Chi² = 11.21, df = 7 (*p =* 0.13); I² = 38%							Test for overall effect: Z = 0.95 (*p* = 0.34)

Abbreviations: IL, Interleukin; OR, Odds ratio; CI, Confidence interval. All models were analyzed based on a random-effects model, except “AA vs. TT + TA,” which was based on a fixed-effects model.

**Table 5 medicina-57-00405-t005:** Forest plot analysis of the association between *IL-6 (-174G/C)* polymorphism and oral cancer risk based on five genetic models.

Genetic Model	First Author, Publication Year	Case		Control		Weight	Odds Ratio
		Events	Total	Events	Total		M-H, Random, 95%CI
C vs. G	Vairaktaris, 2008	138	324	72	312	25.5%	2.47 [1.75, 3.49]
	Gaur, 2011	49	280	69	240	24.9%	0.53 [0.35, 0.80]
	Singh, 2015	143	544	65	370	25.6%	1.67 [1.20, 2.32]
	Fernández-Mateos, 2019	83	140	101	140	24.0%	0.56 [0.34, 0.93]
Subtotal (95%CI)			1288		1062	100.0%	1.07 [0.50, 2.26]
Total events		413		307			
Heterogeneity: Tau² = 0.54; Chi² = 44.63, df = 3 (*P* < 0.00001); I² = 93%							Test for overall effect: Z = 0.17 (*p* = 0.87)
CC vs. GG	Vairaktaris, 2008	18	60	6	96	24.7%	6.43 [2.38, 17.37]
	Gaur, 2011	7	105	14	79	24.9%	0.33 [0.13, 0.87]
	Singh, 2015	21	171	9	138	25.9%	2.01 [0.89, 4.54]
	Fernández-Mateos, 2019	25	37	39	47	24.5%	0.43 [0.15, 1.19]
Subtotal (95%CI)			373		360	100.0%	1.17 [0.31, 4.36]
Total events		71		68			
Heterogeneity: Tau² = 1.57; Chi² = 23.28, df = 3 (*P* < 0.0001); I² = 87%							Test for overall effect: Z = 0.23 (*p* = 0.82)
GC vs. GG	Vairaktaris, 2008	102	144	60	150	26.7%	3.64 [2.24, 5.92]
	Gaur, 2011	35	130	41	106	26.0%	0.58 [0.34, 1.01]
	Singh, 2015	101	251	47	176	27.4%	1.85 [1.22, 2.81]
	Fernández-Mateos, 2019	33	45	23	31	20.0%	0.96 [0.34, 2.71]
Subtotal (95% CI)			570		463	100.0%	1.44 [0.64, 3.26]
Total events		271		171			
Heterogeneity: Tau² = 0.59; Chi² = 25.23, df = 3 (*P* < 0.0001); I² = 88%							Test for overall effect: Z = 0.87 (*p* = 0.38)
CC + GC vs. GG	Vairaktaris, 2008	120	162	66	156	26.1%	3.90 [2.43, 6.26]
	Gaur, 2011	42	140	55	120	25.8%	0.51 [0.30, 0.84]
	Singh, 2015	122	272	56	185	26.6%	1.87 [1.26, 2.78]
	Fernández-Mateos, 2019	58	70	62	70	21.6%	0.62 [0.24, 1.63]
Subtotal (95%CI)			644		531	100.0%	1.28 [0.50, 3.26]
Total events		342		239			
Heterogeneity: Tau² = 0.82; Chi² = 37.38, df = 3 (*P* < 0.00001); I² = 92%							Test for overall effect: Z = 0.51 (*p* = 0.61)
CC vs. GG + GC	Vairaktaris, 2008	18	162	6	156	23.8%	3.13 [1.21, 8.09]
	Gaur, 2011	7	140	14	120	23.9%	0.40 [0.16, 1.02]
	Singh, 2015	21	272	9	185	25.5%	1.64 [0.73, 3.66]
	Fernández-Mateos, 2019	25	70	39	70	26.9%	0.44 [0.22, 0.87]
Subtotal (95%CI)			644		531	100.0%	0.96 [0.37, 2.50]
Total events		71		68			
Heterogeneity: Tau² = 0.77; Chi² = 15.80, df = 3 (*P* = 0.001); I² = 81%							Test for overall effect: Z = 0.09 (*p* = 0.93)

Abbreviations: IL, Interleukin; OR, Odds ratio; CI, Confidence interval. All models were analyzed based on a random-effects model.

**Table 6 medicina-57-00405-t006:** Subgroup analysis of the association between *IL-8 (-251T/A)* polymorphism and oral cancer susceptibility.

Subgroups (N)	A vs. T				AA vs. TT				TA vs. TT				AA + TA vs. TT				AA vs. TT + TA			
	OR	95%CI	*P*	P_h_	OR	95%CI	*P*	P_h_	OR	95%CI	*P*	P_h_	OR	95%CI	*P*	P_h_	OR	95%CI	*P*	P_h_
Overall (8)	0.97	[0.76, 1.23]	0.78	<0.0001	0.86	[0.53, 1.41]	0.55	0.001	0.78	[0.46, 1.33]	0.37	<0.00001	0.83	[0.51, 1.35]	0.45	<0.00001	1.10	[0.90, 1.33]	0.34	0.13
Ethnicity																				
Caucasian (3)	1.11	[0.61, 2.00]	0.73	<0.00001	1.35	[0.31, 5.80]	0.69	<0.00001	069	[0.19, 2.56]	0.58	<0.00001	0.81	[0.23, 2.84]	0.74	<0.00001	1.40	[0.78, 2.50]	0.26	0.05
Asian (4)	0.88	[0.74, 1.06]	0.17	0.73	0.81	[0.56, 1.18]	0.27	0.86	0.85	[0.65, 1.11]	0.23	0.28	0.84	[0.65, 1.08]	0.17	0.42	0.87	[0.62, 1.23]	0.44	0.81
Mixed (1)	0.92	[0.65, 1.30]	0.64	-	0.82	[0.40, 1.65]	0.57	-	1.20	[0.67, 2.14]	0.55	-	1.06	[0.62, 1.84]	0.82	-	0.73	[0.40, 1.32]	0.30	-
Source of control																				
Population-based (7)	0.97	[0.74, 1.26]	0.82	<0.0001	0.87	[0.51, 1.50]	0.62	0.0005	0.76	[0.43, 1.37]	0.36	<0.00001	0.82	[0.48, 1.40]	0.46	<0.00001	1.11	[0.91, 1.35]	0.31	0.09
Hospital-based (1)	0.92	[0.51, 1.65]	0.79	-	0.84	[0.25, 2.79]	0.77	-	0.95	[0.39, 2.32]	0.92	-	0.92	[0.40, 2.13]	0.85	-	0.86	[0.29, 2.58]	0.79	-
Genotyping method																				
PCR-RFLP (3)	1.02	[0.59, 1.78]	0.94	<0.00001	0.94	[0.28, 3.18]	0.92	0.0003	0.71	[0.20, 2.49]	0.59	<0.00001	0.78	[0.24, 2.55]	0.68	<0.00001	1.14	[0.57, 2.28]	0.71	0.02
TaqMan (3)	0.99	[0.83, 1.19]	0.95	0.17	1.04	[0.72, 1.49]	0.84	0.33	0.83	[0.51, 1.35]	0.46	0.11	0.91	[0.68, 1.20]	0.50	0.14	1.10	[0.81, 1.50]	0.53	0.21
Other (2)	0.96	[0.67, 1.39]	0.84	0.85	1.03	[0.46, 2.33]	0.93	0.65	0.86	[0.51, 1.47]	0.59	0.78	0.89	[0.54, 1.48]	0.67	0.93	1.10	[0.51, 2.35]	0.81	0.55

The numbers had no statistically significant values (*p* > 0.05). Abbreviations: OR, odds ratios; 95%CI, 95% confidence interval; SCC, squamous cell carcinoma; PCR-RFLP, polymerase chain reaction-restriction fragment length polymorphism; N, number of studies; P_h_, P_heterogeneity_.

**Table 7 medicina-57-00405-t007:** Subgroup analysis of the association between *IL-6 (-174G/C)* polymorphism and oral cancer susceptibility.

Subgroups (N)	C vs. G				CC vs. GG				GC vs. GG				CC + GC vs. GG				CC vs. GG + GC			
	OR	95%CI	*P*	P_h_	OR	95%CI	*P*	P_h_	OR	95%CI	*P*	P_h_	OR	95%CI	*P*	P_h_	OR	95%CI	*P*	P_h_
Overall (4)	1.07	[0.50, 2.26]	0.87	<0.00001	1.17	[0.31, 4.36]	0.82	<0.0001	1.44	[0.64, 3.26]	0.38	<0.0001	1.28	[0.50, 3.26]	0.61	<0.00001	0.96	[0.37, 2.50]	0.93	0.001
Ethnicity																				
Caucasian (3)	1.07	[0.50, 2.26]	0.87	<0.00001	1.17	[0.31, 4.36]	0.82	<0.0001	1.44	[0.64, 3.26]	0.38	<0.0001	1.28	[0.50, 3.26]	0.61	<0.00001	0.96	[0.37, 2.50]	0.93	0.001
Source of control																				
Population-based (2)	2.03	[1.38, 2.97]	**0.0003**	0.11	0.96	[0.21, 4.35]	0.95	0.02	2.56	[1.32, 4.99]	**0.005**	0.04	2.67	[1.30, 5.47]	**0.007**	0.02	0.96	[0.37, 2.50]	0.93	0.001
Hospital-based (2)	0.54	[0.39, 0.74]	**0.0002**	0.84	1.46	[0.08, 26.60]	0.80	<0.0001	0.65	[0.40, 1.06]	0.08	0.41	0.53	[0.34, 0.83]	**0.006**	0.71	0.43	[0.25, 0.74]	**0.002**	0.86
Genotyping method																				
PCR-RFLP (3)	1.31	[0.56, 3.04]	0.54	<0.00001	0.68	[0.21, 2.19]	0.51	0.009	1.59	[0.61, 4.19]	0.35	<0.00001	1.55	[0.53, 4.59]	0.43	<0.00001	1.27	[0.41, 3.97]	0.68	0.008
TaqMan (1)	0.56	[0.34, 0.93]	**0.02**	-	6.43	[2.38, 17.37]	**0.0002**	-	0.96	[0.34, 2.71]	0.93	-	0.62	[0.24, 1.63]	0.34	-	0.44	[0.22, 0.87]	**0.02**	-

Bold numbers represent statistically significant values (*p* < 0.05). Abbreviations: OR, odds ratios; 95% CI, 95% confidence interval; SCC, squamous cell carcinoma; PCR-RFLP, polymerase chain reaction-restriction fragment length polymorphism; N, number of studies; P_h_, P_heterogeneity_.

**Table 8 medicina-57-00405-t008:** Meta-regression analysis based on two variables for the association between *IL-8 (-251T/A)* and *IL-6 (-174G/C)* polymorphisms and oral cancer susceptibility.

Polymorphism	Variable	Allele Model			Homozygote Model			Heterozygote Model			Recessive Model			Dominant Model		
		R	Adjusted R^2^	*P*	R	Adjusted R^2^	*P*	R	Adjusted R^2^	*P*	R	Adjusted R^2^	*P*	R	Adjusted R^2^	*P*
*IL-8 (-251T/A)*	Year of publication	0.484	0.106	0.225	0.348	−0.025	0.398	0.239	−0.100	0.569	0.374	−0.003	0.361	0.351	−0.023	0.395
*IL-6 (-174G/C)*		0.594	0.030	0.406	0.913	0.751	0.087	0.595	0.032	0.405	0.653	0.140	0.347	0.645	0.125	0.355
*IL-8 (-251T/A)*	Number of participants	0.022	−0.166	0.959	0.122	−0.149	0.773	0.111	−0.152	0.794	0.069	−0.161	0.872	0.307	−0.057	0.460
*IL-6 (-174G/C)*		0.616	0.069	0.384	0.585	0.014	0.415	0.414	−0.243	0.586	0.462	−0.180	0.538	0.516	−0.100	0.484

R, correlation coefficient. The numbers had no statistically significant values (*p* > 0.05).

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
