# Peer review of "Association between IL-8 (-251T/A) and IL-6 (-174G/C) Polymorphisms and Oral Cancer Susceptibility: A Systematic Review and Meta-Analysis"

_medicina, 2021, doi:10.3390/medicina57050405_

Round 1

Reviewer 1 Report

Manuscript may be accepted for publication. 

Author Response

We thank Reviewer #1 once again for the care devoted to the revised manuscript. 

Reviewer 2 Report

I am satisfied with the update the authors made to the manuscript based on my recommendation. The manuscript looks scientifically sounder now and can be considered for publication! 

Author Response

We thank Reviewer 2 once again for the care devoted to the revised manuscript. 

This manuscript is a resubmission of an earlier submission. The following is a list of the peer review reports and author responses from that submission.

Round 1

Reviewer 1 Report

The manuscript is dealing with problems of detection of oral cancer. Topic of cytokines and disease developement is well known, however presented work significantly enhances this state of the art.

Author Response

Thank you very much for the care devoted to thoroughly review the manuscript. Please find the detailed point-by-point-response attached as a separate file. Again, thank you very much for all your kind efforts.

Reviewer 2 Report

The present article evaluated  a possible association between IL-8 (-251T/A) and IL-6 (-174G/C) polymorphisms and oral cancer susceptibility. The manuscript is of interest, however some improvements should be needed firstly to consider it for publication:
- In "data source and literature search", please correct the sentence: "The used search terms used";
- A PICO question should be included in results;
- A value of k-statistic should be reported;
- Trial sequential analysis would strongly improve the quality of the manuscript by quantifying the power of the available evidence.

Author Response

(The authors gave the same response as above.)

Reviewer 3 Report

The authors performed an interesting metadata study where they examined the role of cytokines IL-8 and IL-6 and their polymorphism on Oral cancer risk and included data from all the available published articles to examine. 

Here are my comments regarding their study –

Devising and using the right search string is very important in the kind of research authors performed. The search string the authors used to look for the articles is not complete. They used (“oral cancer” or “oral carcinoma” or “oral cavity” or “oral squamous cell carcinoma” or “oral SCC” or “OSCC” or “tongue cancer” or “oropharyngeal squamous cell carcinoma” or “oropharynx cancer”) as the search terms for Oral cancer. However, when we look in the standard NCBI database called “Mesh” which is an NLM controlled vocabulary thesaurus used for indexing articles for PubMed, we can find that there are lots of other synonyms for Oral Cancer with which articles are indexed in Pubmed apart from what authors in this article used. For example – “Mouth Neoplasm”, “Oral Neoplasm*”, “Mouth Cancers” etc. There were more than 15 synonyms of Oral cancers mentioned in Mesh database, and knowing how Pubmed search engine works, authors may have missed a plethora of articles as they did not use many synonyms for Oral Cancers in their search string that they mentioned.” 

The authors claim to look through all the available articles systematically and they have mentioned a nice layout of their article screening process in Figure 1. However, missing to use all the available synonyms which I mentioned above creates significant doubt in their claim. Since literature search, review, and later data extraction from these literature forms the basis of their examination, I cannot trust the conclusions given by the authors because they may have missed many articles if those articles used the search terms that the authors did not use in their search string that they devised.

The authors should provide a detailed explanation why they did not use all the available synonyms for “Oral Cancer,” which they can have easily found in the Mesh database of Pubmed, and how missing lots of synonyms of “Oral cancer” might or might not impact their conclusions they made. 

Author Response

(The authors gave the same response as above.)
